# LPCF-YOLO: A YOLO-Based Lightweight Algorithm for Pedestrian Anomaly Detection with Parallel Cross-Fusion

**DOI:** 10.3390/s25092752

**Published:** 2025-04-26

**Authors:** Peiyi Jia, Hu Sheng, Shijie Jia

**Affiliations:** School of Rail Intelligent Engineering, Dalian Jiaotong University, Dalian 116028, China; jiapy@djtu.edu.cn (P.J.); hu.sheng@djtu.edu.cn (H.S.)

**Keywords:** LPCF-YOLO, lightweight feature extraction, pedestrian abnormal detection, parallel cross-fusion

## Abstract

To address the issue of high complexity in current pedestrian anomaly detection network models, which hinders real-world deployment, this paper proposes a lightweight anomaly detection network called LPCF-YOLO (Lightweight Parallel Cross-Fusion YOLO) based on the YOLOv8n model. Firstly, the FPC-F (Fast Parallel Cross-Fusion) module, which incorporates PConv, and the S-EMCP (Space-efficient Merging Convolution Pooling) module are designed in the backbone network to replace C2F and SPPF at various scale branches. Additionally, an ADown module is introduced in the third layer to reduce the computational cost. In the neck network, a Lightweight High-level Screening Feature Pyramid Network (L-HSFPN) is designed to replace the PAFPN structure. Furthermore, the Wise-IoU loss function is employed to enhance the model’s localization performance and generalization ability. The experimental results in the UCSD-Ped1 and UCSD-Ped2 datasets show that, compared to YOLOv8n, the proposed approach reduces parameters by 30.33% and FLOPs by 79.01%, achieving 2.09 M parameters and 1.7 G FLOPs; it also results in a 179.62% increase in FPS to 43.9. Meanwhile, the mean average precision (mAP@0.5) is either maintained (in the UCSD-Ped2 dataset) or slightly improved (in the UCSD-Ped1 dataset).

## 1. Introduction

Pedestrian anomaly behavior generally refers to actions or movements that deviate from what is considered normal or expected within a specific context [1,2]. Due to the rapid pace of urbanization and increasing surveillance demands, the detection of anomaly behavior has found widespread application in fields such as intelligent surveillance and traffic management. The real-time, accurate detection and recognition of anomalous behavior in crowds has emerged as a pivotal challenge in ensuring public safety [3,4]. Research into efficient and lightweight pedestrian anomaly detection algorithms can enable large-scale deployment in resource-constrained environments, thereby offering significant practical value.

Traditional pedestrian anomaly behavior detection methods typically adhere to a two-stage process: feature extraction (such as trajectory features [5], a Histogram of Oriented Gradients (HOG) [6], and Histogram of Optical Flow (HOF) [7]) and classification (including techniques like K-means clustering, Support Vector Machines (SVM), and Principal Component Analysis (PCA) [8,9]). However, these methods encounter limitations when dealing with complex surveillance scenarios, as the design of feature descriptors often necessitates prior knowledge, which poses difficulties in accommodating a wide variety of anomaly behavior patterns.

Unlike traditional machine learning methods, deep learning automatically learns complex feature representations, thereby circumventing the limitations associated with manually designed features and demonstrating exceptional performance in handling high-dimensional, nonlinear data [10]. Convolutional Neural Networks (CNNs) and Long Short-Term Memory (LSTM) networks effectively capture spatial and temporal dependencies, rendering them highly adaptable to anomaly behavior detection tasks [11]. When trained on large-scale datasets, deep learning techniques exhibit robust generalization capabilities, enabling the detection of anomalous behavior events within complex patterns [12]. Despite their high computational demands, recent advancements in optimization algorithms and hardware technology have made these methods increasingly practical and applicable.

Deep learning-based anomaly detection can be categorized into three primary types: supervised, semi-supervised, and unsupervised. Unsupervised anomaly detection involves extracting patterns and information from unlabeled data to identify anomalies without requiring labeled data [13,14]. Semi-supervised anomaly detection, which incorporates a small set of labeled normal samples along with a large set of unlabeled data, significantly improves the detection accuracy [15,16]. However, it necessitates maintaining a balance between the labeled and unlabeled data, thereby increasing the model complexity.

Supervised learning trains models using input data and corresponding labels to learn the mapping between inputs and outputs. Compared to unsupervised and semi-supervised learning, supervised learning typically attains higher accuracy because it utilizes well-defined labeled data and has clear optimization objectives. Supervised object detection algorithms based on deep learning are mainly categorized into two approaches: two-stage and single-stage detection methods. The two-stage approach, exemplified by the R-CNN series, notably, Faster R-CNN [17], first generates candidate regions and then performs a fine-grained classification of these regions. Single-stage detection methods directly regress the target’s location and category, including approaches such as the SSD (Single-Shot MultiBox Detector) [18,19] and YOLO [20,21,22,23,24,25] series.

Recently, emerging techniques such as Visual Transformers [26] and DETR [27] have advanced object detection, thereby contributing to the progress in anomaly detection as well. However, these methods face challenges in achieving high-accuracy anomaly detection due to their large network architectures, numerous training parameters, and high computational costs. This complexity restricts their deployment on edge devices with limited computational resources and stringent power consumption constraints, consequently impeding the implementation of real-time online anomaly detection.

As one of the most effective single-stage methods, the YOLO (You Only Look Once) algorithm [20,21,22,23,24,25] garners considerable attention for its significant speed advantage, making it highly suitable for real-time surveillance systems. According to the literature [28], abnormal human behavior is treated as an object classification task, where the YOLO object detection model is used to detect and localize anomalies, ensuring both real-time identification and accuracy. Furthermore, the main advantage of YOLO lies in its ability to process an entire image in a single pass, enabling the quick identification of pedestrian anomalies, which is particularly ideal for crowded monitoring environments.

Among the YOLO models, YOLOv8 [22], developed by Ultralytics, offers five model variants—n, s, m, l, and x—tailored to various application scenarios and hardware specifications. As the lightest variant, YOLOv8n provides a foundation for efficient deployment. Based on this foundation, ongoing research focuses on achieving further lightweight optimization while simultaneously enhancing the detection performance.

To address the aforementioned issues, this paper proposes a lightweight anomaly detection network, LPCF-YOLO (Lightweight Parallel Cross-Fusion YOLO), which is based on the YOLOv8n model.

In the backbone network, an FPC-F (Fast Parallel Cross-Fusion) module is developed to replace the C2F module, which is an optimized module that enhances feature extraction by combining ideas from CSP (Cross-Stage Partial Networks) and ELAN (Efficient Layer Aggregation Network) in YOLOv8. The main design ideas are as follows:Lightweight and Efficiency-Oriented.

Using FasterNet’s PConv [29] (selectively processes only a subset of input channels while leaving others untouched) as the computational core, selective channel processing is employed to reduce FLOPs; combining CSPNet’s staged architecture, this approach minimizes the parameter count while maintaining the cross-layer feature interaction.

2.Heterogeneous Feature Synergy Enhancement.

Parallel branches extract local details (PConv processes some channels) and the global context (unprocessed channels retain original information). After fusion, a 1 × 1 convolution is used to achieve channel adaptive weighting, enhancing the expression of key target features.

3.Deployment-Friendly Optimization.

Introducing the ADown module with downsampling strategies dynamically adjusts the computational load (e.g., compressing a 160 × 160 feature map to 80 × 80), ensuring a real-time performance on edge devices.

To address the issue of feature loss that occurs during downsampling in traditional SPPF (Spatial Pyramid Pooling Fast) modules, the S-EMCP (Space-efficient Merging Convolution Pooling) module is designed at the final layer of the backbone, combining max pooling, average pooling, and 1 × 1 convolution in parallel. It enhances the efficiency of extracting deep-level target features through a multi-path feature fusion strategy.

In the neck network, inspired by [30], this paper introduces an L-HSFPN (Lightweight High-level Screening Feature Pyramid Network) to enhance the model performance in multi-scale object detection. L-HSFPN uses lightweight modules (CSPPC) combined with a high-level feature-guided fusion strategy to enhance multi-scale detection capabilities while reducing the computational load.

YOLOv8 utilizes CIoU Loss [31] as the bounding box regression loss function, representing aspect ratios as relative values. However, discrepancies in the orientation between predicted boxes and ground truth boxes introduce oscillations in the predicted box positions during training, which slows convergence and reduces the prediction accuracy. This paper selects Wise-IoU Loss [32] as the bounding box regression loss function. By employing a dynamic, non-monotonic mechanism to design a more rational gradient gain allocation strategy, Wise-IoU Loss effectively reduces gradient gains, thereby enhancing the model’s localization accuracy and generalization capability.

In summary, the main contributions of this paper are as follows:In the backbone network, an FPC-F (Fast Parallel Cross-Fusion) module based on PConv and a S-EMCP (Space-efficient Merging Convolution Pooling) module have been designed to replace the original C2F and SPPF modules, respectively. Additionally, an ADown module has been introduced at the third layer to reduce floating-point computations.In the neck network, the CSPPC module has been designed to replace the C2F module in HSFPN, thus forming the L-HSFPN (Lightweight High-level Screening-feature Pyramid) module.In the bounding box loss function, the Wise-IoU Loss is utilized to replace the CIoU Loss.

The experimental results indicate that, compared to YOLOv8n, LPCF-YOLO reduces parameters and floating-point operations (FLOPs) by 30.33% and 79.01%, respectively. In the UCSD-Ped1 dataset, the mean average precision (mAP@0.5) of LPCF-YOLO improves by 0.25%, while it remains nearly the same in the UCSD-Ped2 dataset. These results demonstrate that LPCF-YOLO effectively reduces the model complexity and computational load while maintaining the detection accuracy.

## 2. Related Work

Compared to the two-stage R-CNN series, the YOLO series achieves fast and accurate object detection through a unified model, eliminating the region proposal and feature extraction stages, thereby enhancing the detection speed and enabling real-time detection. Li et al. [33] proposed an anomaly detection algorithm based on the YOLO network and Conv-AE, utilizing Conv-AE as the foundational network to extract spatial features from video frames. To address interference within the network, they introduced a weighted loss function based on YOLO detection results to reduce the impact of background elements. Li et al. [34] further developed an innovative pedestrian anomaly behavior detection model (PABDM) using a multi-scale fusion YOLOv3 algorithm to identify crowd behavior in abnormal scenarios, significantly improving the detection accuracy and performance. Yuan et al. [35] enhanced YOLOv5 by incorporating SRGAN for input image reconstruction, utilizing MnasNet as the backbone for feature extraction, introducing an ECA-Net attention mechanism within the feature fusion network, and optimizing the loss function with EIoU. These enhancements collectively improved the anomaly detection accuracy and significantly boosted the real-time detection performance. Ganagavalli et al. [36] introduced an automated anomaly detection system based on YOLO, which efficiently detected abnormal behaviors and employed an LSTM-CNN model to select additional features for classification. The authors of [37] presented a novel hybrid learning architecture named UYOLO (U-shape You Only Look Once) which integrates the CSP-based backbone and detection branch to produce three scale high-level features for the object detection around the high-speed railroad. Kong et al. [38] introduced the Drone-DETR model, which is based on RT-DETR to overcome the difficulties associated with detecting small objects and reducing redundant computations arising from complex backgrounds in ultra-wide-angle images.

Although single-stage detectors have achieved significant advancements in their real-time performance, anomaly detection applications continue to face numerous challenges, particularly in feature extraction, feature fusion, and deployment on edge computing platforms such as surveillance cameras or mobile devices. To address these challenges, research primarily focuses on two directions: model lightweighting and the enhancement of the object detection performance.

Currently, mainstream lightweight neural networks like MobileNets [39], ShuffleNets [40], and GhostNet [41] utilize depthwise separable convolutions (DWConv) and grouped convolutions (GConv) for spatial feature extraction. However, to reduce FLOPs, these operations often increase memory access, which can have unintended side effects. Therefore, FasterNet [29] employs Partial Convolution (PConv), which applies filters to only a subset of input channels while leaving the remaining channels unprocessed, thereby achieving rapid and efficient processing. The FLOPs of PConv are lower than those of conventional convolutions, but higher than DWConv’s and GConv’s. Partial Convolution (PConv) reduces the number of parameters and computational complexity, enabling model lightweighting while maintaining a satisfactory performance.

To enhance the detection performance in complex scenes, researchers propose various optimizations in the neck network, such as the Adaptive Feature Pyramid Network (AFPN) [42], Bidirectional Feature Pyramid Network (BiFPN) [43], and High-level Screening Feature Pyramid Network (HSFPN) [30]. These methods improve the detection accuracy through feature fusion and weighting mechanisms. In particular, HSFPN employs a channel attention module, using high-level feature maps as weighted filters for lower-level features. These filtered features are then combined with high-level features, enriching the semantic information of lower-level features and enhancing the network’s feature extraction capabilities. Similarly, these advancements in neck networks, combined with the efficiency of PConv, contribute to its popularity and widespread application.

The evolution of loss functions from simple forms gradually improves the object detection accuracy. Early bounding box loss functions, such as CIoU Loss [31] and EIoU Loss [44], primarily rely on Intersection over Union (IoU) to measure the overlap between the anchor boxes and target boxes. Although these loss functions achieve a balance between the localization accuracy and regression efficiency, they still encounter issues such as vanishing gradients when handling small objects or non-overlapping bounding boxes. To address these limitations, SIoU Loss [45] introduces additional geometric factors, such as the angle and distance, further enhancing the accuracy of bounding box regression. However, SIoU continues to assign excessive weight to high-quality anchor boxes, causing the model to over-focus on a few anchors. To overcome this, MPDLoU [46] and Wise-IoU [32] introduce further optimizations, with Wise-IoU incorporating a dynamic, non-monotonic focusing mechanism (FM) that takes into account anchor outliers in addition to IoU. This mechanism distributes gradient gains more appropriately, reducing competition among high-quality anchors and effectively suppressing harmful gradient interference from low-quality samples. Consequently, Wise-IoU substantially boosts the effectiveness of gradient backpropagation and elevates the overall performance of the detector.

## 3. Methodology

The overall framework of LPCF-YOLO is illustrated in Figure 1. The backbone network (LPCF-YOLO-Backbone) incorporates a newly designed FPC-F (Fast Parallel Cross-Fusion) module for feature extraction and integrates an ADown module after the second feature map module. Additionally, an S-EMCP (Space-efficient Merging Convolution Pooling) module is introduced in the final layer of the backbone. This series of optimizations enhances the model’s feature extraction efficiency, computational performance, and real-time processing capabilities. Meanwhile, the L-HSFPN (Lightweight High-level Screening Feature Pyramid Network) replaces the original PANet (Path Aggregation Network) in YOLOv8 as the neck network, effectively aggregating features from all backbone layers to strengthen multi-scale feature representation. Furthermore, Wise-IoU Loss replaces the CIoU Loss used in YOLOv8, employing an anchor-free detection approach to directly predict the target’s center and aspect ratio. This approach not only reduces the computational complexity, but also improves the model’s accuracy.

### 3.1. Backbone

The LPCF-YOLO-Backbone processes input data with a resolution of 640×640 pixels. Initially, two 3 × 3 convolutions with a stride of 2 are applied, resulting in a downsampled feature map of 160 × 160 pixels. Subsequently, the ADown module further downsamples the feature map to 80 × 80. The P3 detection layer then applies a 3 × 3 convolution with a stride of 2, enhanced by an FPC-F module to improve the model’s perceptual capability. Similarly, the P4 detection layer incorporates another 3 × 3 convolution with a stride of 2 and a second FPC-F module, further downsampling the feature map to 20 × 20 pixels.

Finally, the P5 detection layer combines a 3 × 3 convolution (with a stride of 2), an FPC-F module, and an S-EMCP module, effectively promoting the high-precision detection of small objects.

To improve the detection accuracy, we propose the Fast Parallel Cross-Fusion (FPC-F) module, drawing inspiration from the Cross-Stage Partial Network (CSPNet) and FasterNet. As illustrated in Figure 2, the FPC-F module adopts a three-branch structure for feature extraction and fusion, enabling efficient representation learning. Specifically, it applies partial convolution (PConv) to only 1/8 of the input channels, while the remaining 7/8 bypass the transformation, reducing the computational complexity while preserving essential information. The extracted features are then aggregated through a 1 × 1 convolution, ensuring effective information fusion across different scales. By integrating shallow and deep features more efficiently, FPC-F enhances information transmission and utilization, ultimately leading to an improved detection performance.

As illustrated in Figure 3, the Partial Convolution (PConv) method divides the input channels into two parts: one part undergoes a 3 × 3 convolution, while the other part remains unchanged and is subsequently appended to the feature channels. In PConv, a split-and-concatenate operation (split_cat) recombines the convolved and unprocessed channels. Within the FPC-F module, PConv applies convolution to 1/8 of the input channels, leaving the remaining 7/8 channels unchanged.

The S-EMCP module adopts a multi-path feature extraction strategy that combines convolutional and pooling operations to improve feature fusion. As illustrated in Figure 2, the input feature map undergoes a 1 × 1 convolution and is then processed by three parallel branches. The first branch retains the original features without modification. The second branch applies 5 × 5 average pooling (stride 1, padding 2) followed by 5 × 5 max pooling (stride 1, padding 2) to capture contextual information. These pooled features are then concatenated along the channel dimension and undergo a 1 × 1 convolution to restore the original channel count. The third branch, in contrast, applies 5 × 5 max pooling followed by 5 × 5 average pooling, concatenates the outputs, and then uses a 1 × 1 convolution to halve the number of channels. Finally, all three branches’ outputs are concatenated and passed through a final convolutional layer, ensuring effective feature fusion and enhanced spatial feature representation.

The ADown module, as illustrated in Figure 2, incorporated in the third layer of the backbone network, serves as an adaptive downsampling mechanism that combines average pooling and max pooling to optimize feature retention while reducing computational complexity. Unlike conventional stride-based convolutional downsampling, ADown ensures that both local and global information is retained more effectively during feature compression. Additionally, it employs dynamic kernel-based adaptive padding, which allows it to flexibly adjust to different input sizes and network configurations. This approach not only preserves critical spatial details during downsampling, but also enhances the overall robustness of the backbone network, making it well-suited for lightweight real-time detection applications.

### 3.2. Neck

The L-HSFPN (Lightweight High-level Screening Feature Pyramid Network) is built upon the HSFPN (High-level Screening Feature Pyramid Network), where the CSPPC (Cross-Stage Partial Parallel Convolution) module replaces the C2f module. In the LPCF-YOLO architecture, input images fed into L-HSFPN are adjusted to three sizes: 40 × 40, 20 × 20, and 10 × 10. After feature selection and extraction, the outputs are unified into 256 channels and passed to the next layer, where feature fusion and detection tasks are completed.

The HSFPN (High-level Screening Feature Pyramid Network) is a structure introduced in the Multi-level Feature Fusion and Deformable Self-attention DETR (MFDS-DETR) [30]. It comprises two main components: a feature selection module and a feature fusion module. The feature selection module first applies global average pooling and global max pooling to process multi-scale feature maps and employs a channel attention (CA) mechanism to weight the feature channels, ensuring that the extracted features are more representative. The dimension matching module then utilizes a 1 × 1 convolution to standardize the channel count across different scale feature maps, providing consistent input for feature fusion. The feature fusion module utilizes a selective feature fusion (SFF) mechanism, employing high-level features as weights to filter low-level features. It adjusts the scale of high-level features to match those of lower-level features through a combination of deconvolution and bilinear interpolation. By employing multi-level feature selection and fusion, HSFPN retains high-level semantic information while improving the detection of fine-grained image details, significantly enhancing the overall model performance.

As shown in Figure 4, the CSPPC module is a lightweight architecture based on PConv. Following an initial 1 × 1 convolution, the feature map channels are divided into two parts. One part undergoes feature extraction through two PConvs, with each PConv applying a convolution to 1/4 of the input channels and leaving the remaining 3/4 unchanged. Finally, the two parts are concatenated and processed through another 1 × 1 convolution to produce the output.

### 3.3. Wise-IoU

The Wise-IoU (WIoU) loss function [32] is a dynamic, non-monotonic IoU-based loss that employs a non-monotonic focusing mechanism to assess the anchor box quality based on the “outlier degree (A)” rather than IoU alone. This approach provides a rational gradient gain allocation strategy that enhances the overall performance of the detector. Currently, WIoU exists in three versions: v1, v2, and v3. LPCF-YOLO, in particular, adopts the latest version, WIoU v3.(1)$$L_{\mathrm{WIoUv}3}=rL_{\mathrm{WIoUv}1}\;\;r=\frac{\beta }{\delta \alpha^{\beta -\delta }}$$

WIoU v3 introduces a non-monotonic focusing mechanism to further optimize the gradient allocation for anchor boxes. The WIoU v3 loss function is expressed in Equation (1), where WIoU v3 defines an outlier degree (β) to evaluate the anchor box quality. High-quality anchor boxes have smaller β values, while low-quality anchors have larger β values. The parameter α controls the curvature of the focusing mechanism. A value of 1.8 balances the attention between common and outlier samples. Also, δ is set to a typical value of 3.

The core of WIoU v3 lies in dynamically adjusting the gradient gain for anchor boxes. Specifically, WIoU v3 assigns a smaller gradient gain to high-quality anchors to prevent overfocusing on them, thereby allowing the model to focus more on anchors of a standard quality. For low-quality anchors, WIoU v3 reduces the gradient gain in order to minimize the adverse effects of these samples on model training. This dynamic adjustment mechanism enables WIoU v3 to achieve a more rational gradient distribution, dynamically handling anchors of varying quality during training, ultimately improving the model’s localization accuracy and robustness.

## 4. Experiments and Results

### 4.1. Dataset

The UCSD dataset, developed by the University of California, San Diego, is a publicly available resource widely used in anomaly detection and traffic monitoring research. Under standard conditions, the scenes depict only pedestrians walking on designated pathways, while anomalous events include atypical pedestrian motion patterns or individuals traversing non-pedestrian areas. Anomalies in this dataset are categorized into six types (as shown in Table 1): bike, through the lawn, skateboard, cart, wheelchair, and pushcart. The UCSD dataset is divided into two subsets, Ped1 and Ped2, each representing a distinct scene environment.

To address the presence of numerous non-anomalous frames in the dataset videos, preprocessing is performed to extract all anomalous frames, which are then labeled in the YOLO format using the LabelImg software. After adjustment, the Ped1 subset contains a total of 6243 images, including background images without anomalies and six types of anomalies: bicycles, pedestrians crossing grassy areas, skateboards, pushcarts, wheelchairs, and cars, as shown in Figure 5. The Ped1 dataset is randomly divided into training, testing, and validation sets in a 7:2:1 ratio. The filtered and labeled Ped2 subset includes 2118 images, comprising background images with no anomalies and three types of anomalies: bicycles, skateboards, and cars, as illustrated in Figure 6. The Ped2 dataset is similarly split into training, testing, and validation sets in a 7:2:1 ratio.

### 4.2. Experimental Environment and Training Parameter Settings

The experimental environment is presented in Table 2. The input image size is set to 640 × 640 pixels, with a batch size of 16 and a total of 250 training epochs. The initial learning rate (Lr0) is configured to 0.01 and gradually decreases to a final learning rate (Lrf) of 0.0005. The momentum is set to 0.937, and the weight decay is set to 0.0005.

### 4.3. Evaluation Indicators

The performance evaluation metrics selected for the experiment include precision, recall, and the mean Average Precision (mAP), which are shown in Formulas (2)–(4).

Precision (P) represents the proportion of correctly predicted samples within the positive sample set, where TP denotes the number of true positives, FP represents the number of false positives, and FN stands for the number of false negatives.(2)$$P=\frac{TP}{TP+FP}$$

Recall (R) represents the ratio of correctly predicted positive samples to the total number of actual positive samples. It is determined using Equation (3), where FN denotes the objects that are present but remain undetected.(3)$$R=\frac{TP}{TP+FN}$$

AP (Average Precision) represents the average accuracy of the model, while the mAP (mean Average Precision) is the mean of the AP values across all categories. Specifically, n is the number of categories, and AP_i_ denotes the average precision for category i. mAP@0.5 represents the average precision when the IoU threshold is set to 0.5, while mAP@0.5:0.95 represents the average precision over IoU thresholds ranging from 0.5 to 0.95 with a step size of 0.05.(4)$$mAP=\frac{1}{n}\sum_{i=1}^{n}AP_{i}$$

### 4.4. Comparative Experimental Results and Analysis

#### 4.4.1. Comparison Results and Analysis of LPCF-YOLO and SOTA Algorithms

To assess the effectiveness of the proposed model, LPCF-YOLO is rigorously compared to several lightweight YOLO variants using the UCSD-Ped1 and UCSD-Ped2 datasets. All models are evaluated under identical conditions, utilizing default parameter settings and a standardized input resolution of 640 × 640 to ensure experimental fairness and comparability. The performance metrics of YOLOv5n, YOLOv7-tiny, YOLOv8n, YOLOv9t, YOLOv10n, and YOLOv11n are reported in Table 3 and Table 4, forming the basis for the following key conclusions.

Compared to existing YOLO models, LPCF-YOLO exhibits significant advantages in both parameter efficiency and computational complexity, owing to its lightweight architecture that substantially reduces resource demands. Specifically, LPCF-YOLO reduces the parameter count by 16.73% and GFLOPs by 76.06% relative to YOLOv5n, while achieving a comparable mAP@0.5:0.95 of 75.83%, which is 0.22% lower than YOLOv5n’s 76.05%. Similarly, compared to YOLOv8n, LPCF-YOLO reduces the parameters by 30.33% and GFLOPs by 79.01%, achieving a mAP@0.5:0.95 of 75.83%, 0.09% higher than YOLOv8n’s 75.74%.

Although LPCF-YOLO has a slightly higher parameter count than YOLOv9t (2.09 M vs. 2.01 M), it achieves a 77.63% reduction in GFLOPs, indicating superior computational efficiency, while maintaining a mAP@0.5:0.95 of 75.83%, surpassing YOLOv9t’s 74.31% by 1.52%. Furthermore, the recently introduced YOLOv10n and YOLOv11n contain 0.61 M and 0.49 M more parameters, respectively, than LPCF-YOLO, with GFLOPs 4.82 and 3.71 times higher, achieving mAP@0.5:0.95 scores of 75.12% and 77.49%, respectively. These findings highlight LPCF-YOLO’s suitability for resource-constrained environments.

In terms of the detection accuracy, LPCF-YOLO demonstrates a competitive performance. In the UCSD-Ped1 dataset, it achieves a mAP@0.5 of 98.92%, outperforming YOLOv5n, YOLOv7-tiny, and YOLOv9t by 0.05%, 0.39%, and 0.31%, respectively, with particularly strong results in the Skateboard (99.58%) and Cart (99.54%) categories. Compared to YOLOv8n, LPCF-YOLO improves mAP@0.5 by 0.25% (from 98.67% to 98.92%), while maintaining a competitive mAP@0.5:0.95 of 75.83%, which is 0.09% higher than YOLOv8n’s 75.74%. Notably, LPCF-YOLO exhibits a 0.50% improvement in the Skateboard category, 0.16% in the Cart category, and 0.95% in the “Through the Lawn” category. Although YOLOv11n achieves a mAP@0.5 of 98.98% in the UCSD-Ped1 dataset, exceeding LPCF-YOLO by 0.06%, LPCF-YOLO has 0.49 M fewer parameters (2.09 M vs. 2.58 M) and a significantly lower computational complexity (1.7 GFLOPs vs. 6.3 GFLOPs). Additionally, LPCF-YOLO’s mAP@0.5:0.95 of 75.83% remains 1.66% lower than YOLOv11n’s 77.49%, demonstrating an effective balance between the accuracy and computational efficiency.

In the UCSD-Ped2 dataset, LPCF-YOLO achieves a mAP@0.5 of 99.49%, outperforming YOLOv7-tiny by 0.01% and performing comparably to both YOLOv9t and YOLOv10n. While YOLOv5n, YOLOv8n, and YOLOv11n achieve a mAP@0.5 of 99.50%, LPCF-YOLO is only 0.01% lower, while maintaining a lightweight architecture with only 2.09 M parameters and 1.7 GFLOPs, compared to YOLOv8n’s 3.00 M parameters and 8.1 GFLOPs. Moreover, its mAP@0.5:0.95 of 75.83% remains competitive, compared to YOLOv5n (76.05%), YOLOv7-tiny (73.96%), YOLOv8n (75.74%), and YOLOv10n (75.12%).

In summary, LPCF-YOLO achieves an optimal balance between its detection accuracy and computational efficiency. In the UCSD-Ped1 and UCSD-Ped2 datasets, LPCF-YOLO attains mAP@0.5 scores of 98.92% and 99.49%, respectively, while significantly reducing the parameter count (only 2.09 M) and computational complexity (1.7 GFLOPs). Despite its lightweight design, LPCF-YOLO maintains a mAP@0.5:0.95 of 75.83%, which is comparable to the state-of-the-art lightweight YOLO models, further validating its suitability for deployment in embedded and mobile applications.

#### 4.4.2. Comparison and Visualization Analysis of Evaluation Metrics Between LPCF-YOLO and YOLOv8n in the UCSD-Ped1 and UCSD-Ped2 Datasets

To clearly illustrate the performance comparison between LPCF-YOLO and YOLOv8n in the UCSD-Ped1 and UCSD-Ped2 datasets, a visual analysis of mAP@0.5 and mAP@0.5:0.95 on the validation set during training is presented in Figure 7 and Figure 8. As shown in Figure 7, in the UCSD-Ped1 dataset, LPCF-YOLO’s mAP@0.5 gradually approaches YOLOv8n’s after the 125th epoch and surpasses it by the 160th epoch. LPCF-YOLO’s mAP@0.5:0.95 follows a similar trend, demonstrating a continuous performance improvement.

As shown in Figure 8, in the UCSD-Ped2 dataset, although LPCF-YOLO’s mAP@0.5:0.95 is slightly lower than YOLOv8n’s during the first 100 epochs, it shows a notable improvement by the 200th epoch, with stable performance enhancements in subsequent training. Similarly, LPCF-YOLO’s mAP@0.5 on this dataset remains comparable to that of YOLOv8n.

### 4.5. Results and Analysis of Ablation Experiments

#### 4.5.1. The Overall Ablation Experiment of LPCF-YOLO

(1) The LPCF-YOLO-Backbone is essential to the model’s lightweight design and performance enhancement. As shown in Table 5, replacing the backbone with LPCF-YOLO-Backbone increases the mAP@0.5 in the UCSD-Ped1 dataset from 98.67% to 98.71%, an improvement of 0.04%. In the UCSD-Ped2 dataset, the mAP@0.5 slightly decreases from 99.50% to 99.48%, a minimal reduction of 0.02%. Despite this slight decrease in the UCSD-Ped2 dataset, the overall accuracy remains high. Notably, GFLOPs decrease significantly from 8.1 to 2.1, a reduction of 74.07%, and FPS improves from 15.7 to 44.3, an increase of 182.17%. This result indicates that the LPCF-YOLO-Backbone significantly reduces floating-point computations while maintaining accuracy, enhancing the model efficiency in low-resource environments—an essential feature for real-time applications.

(2) The L-HSFPN module primarily enhances feature fusion, improving the detection accuracy while reducing the computational complexity. With this module, the mAP@0.5 in the UCSD-Ped1 dataset increases from 98.67% to 98.87%, an improvement of 0.2%, and in the UCSD-Ped2 dataset, the mAP@0.5 remains high at 99.50%. Additionally, GFLOPs decrease by 17.28%, and the parameter count reduces by approximately 36.67%. These results demonstrate that the L-HSFPN module not only improves the detection accuracy, particularly in the UCSD-Ped1 dataset, but also plays a significant role in reducing the parameter count and computational complexity.

(3) The introduction of the WIoU loss function optimizes the precision of bounding box regression. When combined with the LPCF-YOLO-Backbone and the L-HSFPN module, replacing the loss function with WIoU results in further enhancements in the detection accuracy. Specifically, in the UCSD-Ped1 dataset, the mAP@0.5 increases from 98.64% to 98.92%, an improvement of 0.28%, while in the UCSD-Ped2 dataset, the mAP@0.5 remains stable at 99.50%. These results indicate that WIoU more effectively improves the localization accuracy of high-quality anchor boxes while mitigating the negative gradient impact of low-quality samples, thereby enhancing the model’s overall detection performance.

#### 4.5.2. Ablation Experiment Focused on the Internal Structure of the LPCF-YOLO-Backbone

Ablation experiments are conducted specifically on the LPCF-YOLO-Backbone, with detailed results presented in Table 6. The key observations are as follows.

The FPC-F module effectively enhances the detection accuracy. Integrating the FPC-F module into the backbone network improves the mAP@0.5 in the UCSD-Ped1 dataset from 98.67% to 98.98%, an increase of 0.31%, while maintaining a high mAP@0.5 of 99.50% in the UCSD-Ped2 dataset. By utilizing parallel cross-fusion and partial convolution (PConv), the FPC-F module more effectively aggregates multi-channel information, thereby enhancing the network’s capability in multi-scale feature extraction and fusion. Although the FPC-F module increases the parameter count (from 3.00 M to 3.36 M), it reduces GFLOPs by 0.3 G and improves the FPS from 15.7 to 16.

The S-EMCP module maintains high accuracy while reducing computational complexity. When applied to the final layer of the backbone network, the S-EMCP module increases the mAP@0.5 in the UCSD-Ped1 dataset by 0.11%, and the mAP@0.5 in the UCSD-Ped2 dataset stays at 99.50%. Additionally, this module reduces the parameter count from 3.00 M to 2.84 M and lowers GFLOPs from 8.1 G to 8.0 G. Thus, the S-EMCP module effectively reduces the computational complexity without compromising the detection accuracy.

The ADown module significantly enhances the real-time processing capability. Inserting the ADown module into the third layer of the backbone network reduces GFLOPs by 72.84% and increases FPS by 177.07%, substantially improving the model’s real-time processing performance with a minimal impact on the detection accuracy.

The combined application of multiple modules significantly enhances the overall performance. When both the FPC-F and S-EMCP modules are introduced, the mAP@0.5 in the UCSD-Ped1 dataset improves by 0.12% (from 98.67% to 98.79%), while the mAP@0.5 in the UCSD-Ped2 dataset stays at 99.50%. Further integrating the ADown module on this basis reduces GFLOPs by 72.73% and increases FPS by 170%. This combination achieves substantial reductions in the computational complexity and improvements in the real-time processing capability with a minimal impact on the accuracy.

#### 4.5.3. Comparison of Heatmap Results Across Different Network Layers Between YOLOV8n and LPCF-YOLO

Gradient-weighted Class Activation Mapping (Grad-CAM) generates a heatmap by backpropagating the model’s gradient information to the final convolutional layer, visually highlighting the regions of the image the model focuses on during decision making. In the heatmap, high-gradient regions are shown in red, while low-gradient regions are depicted in blue. The experimental results indicate that the proposed LPCF-YOLO model primarily focuses on the central area of the target, suggesting high accuracy in object localization. Figure 9 compares the heatmaps of the YOLOv8 and LPCF-YOLO models at the 1st, 3rd, 5th, and 7th convolutional layers. The rows in Figure 9 represent YOLOv8 in the first row and LPCF-YOLO in the second, which provides insight into how each model processes features across different convolutional layers and highlighting potential differences in the feature extraction effectiveness.

Figure 9 reveals notable differences in the heatmap representations between LPCF-YOLO and YOLOv8 across various convolutional layers. LPCF-YOLO demonstrates more focused attention across multi-layer convolution outputs, particularly in complex scenes with crowds, accurately concentrating on the central region of the target. Through the FPC-F module and ADown module, shallow feature extraction will focus more on the local details of the target (such as pedestrian limb movements), reducing redundant background interference. The S-EMCP module can preserve the spatial structure of objects in deeper layers, enhancing the global context capture ability for abnormal behaviors (such as cycling and boundary crossing). The L-HSFPN uses a channel attention mechanism to dynamically weight features from different layers. The heatmap shows that it pays more attention to fine-grained features of abnormal behaviors such as skateboarding and pushing carts, resulting in a more accurate classification and localization.

In contrast, YOLOv8’s heatmap appears more dispersed, with high-attention areas lacking focus, which may lead to false positives or missed detections in scenes with complex backgrounds. The shallow heatmaps of YOLOv8n may be more sensitive to background noise (such as lighting variations, occlusions), with more dispersed activation regions. The deep heatmaps of YOLOv8n may suffer from blurred object edges or the omission of small objects.

This comparison indicates that LPCF-YOLO exhibits advantages in feature extraction and multi-level target recognition.

## 5. Conclusions

Building on YOLOv8n, this paper proposes LPCF-YOLO, an anomaly detection algorithm for pedestrians. In the LPCF-YOLO backbone, the FPC-F and S-EMCP modules are integrated to enhance feature representation, enabling the network to capture both fine-grained details and high-level abstract information. This improves its ability to detect targets across various scales. Additionally, the L-HSFPN module is introduced in the neck network, optimizing multi-scale feature fusion while suppressing redundant background noise. The proposed CSPPC module replaces the C2F module in HSFPN, significantly reducing the model complexity. Furthermore, WIoU replaces CIoU as the loss function, mitigating the impact of low-quality samples and enhancing the detection accuracy.

Compared to the baseline YOLOv8n, LPCF-YOLO improves mAP@0.5 by 0.25% in the UCSD-Ped1 dataset while maintaining comparable performance on UCSD-Ped2. Meanwhile, the parameter count is reduced from 3.0 M to 2.09 M, and GFLOPs decrease from 8.1 to 1.7, demonstrating a substantial improvement in the model efficiency.

However, LPCF-YOLO exhibits limitations when applied to datasets with low-resolution frames, occlusions, and limited annotation diversity, which may restrict its generalization ability. Additionally, under extreme lighting conditions (e.g., intense glare or low illumination) and adverse weather (e.g., fog or heavy rain), the detection accuracy may decline. These factors affect the model’s robustness in real-world deployments. Future research will focus on optimizing the network architecture for better efficiency, integrating a dedicated small-object detection layer, and enhancing the model’s scalability across diverse scenarios.

## Figures and Tables

**Figure 1 sensors-25-02752-f001:**
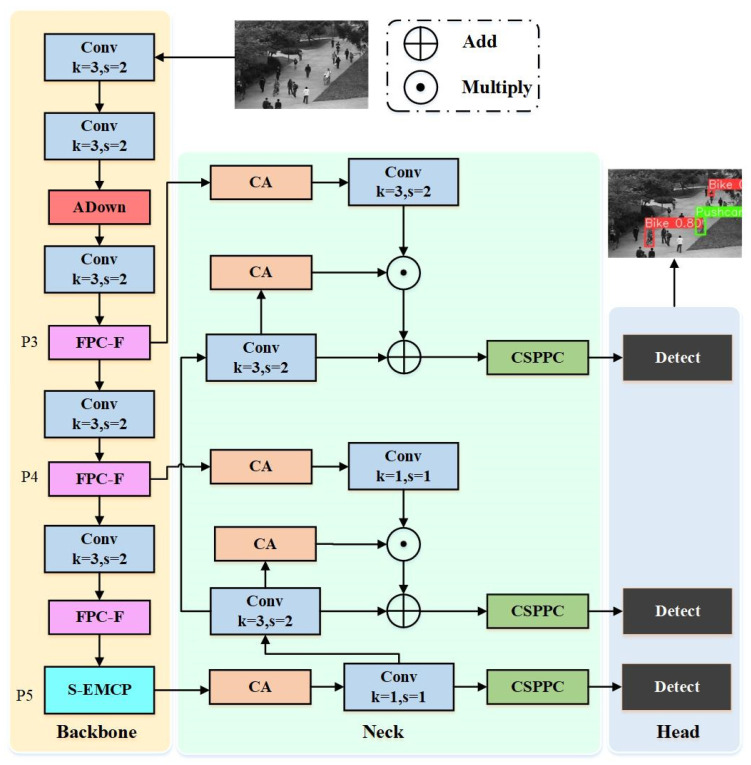
The structure of LPCF-YOLO.

**Figure 2 sensors-25-02752-f002:**
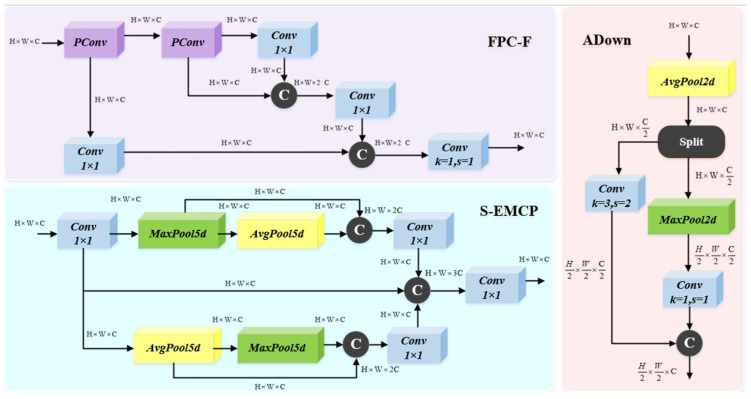
Structural diagrams of the FPC-F module, S-EMCP module, and ADown module.

**Figure 3 sensors-25-02752-f003:**
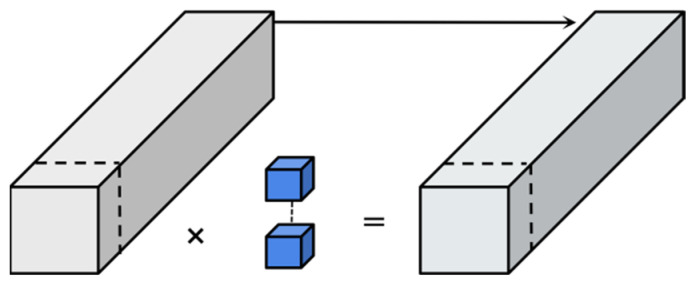
Partial convolution.

**Figure 4 sensors-25-02752-f004:**
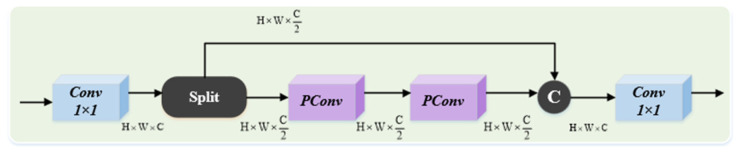
Cross-stage partial parallel convolution.

**Figure 5 sensors-25-02752-f005:**
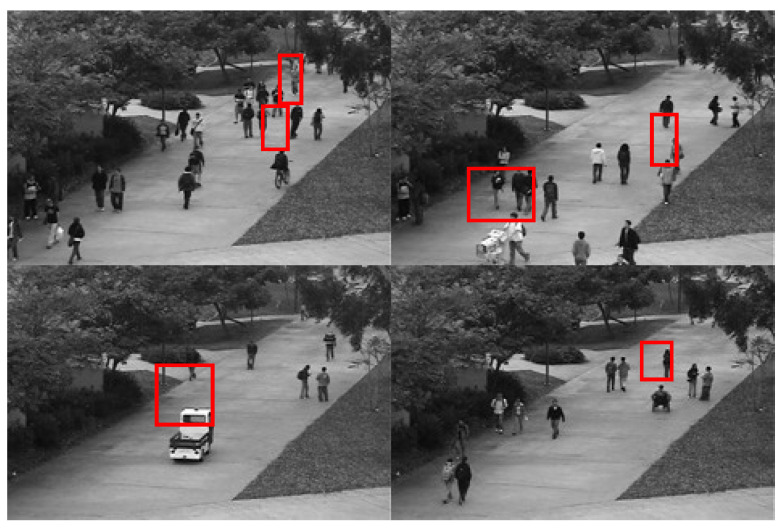
Presentation of categories in the UCSD-Ped1 dataset.

**Figure 6 sensors-25-02752-f006:**
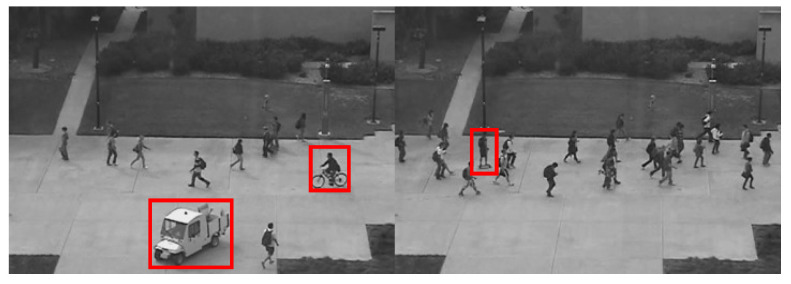
Presentation of categories in the UCSD-Ped2 dataset.

**Figure 7 sensors-25-02752-f007:**
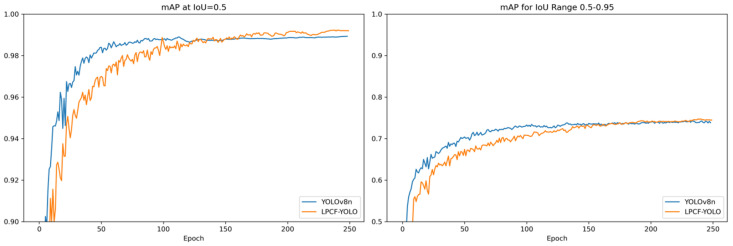
Visualization of mAP@0.5 and mAP@0.5:0.95 metrics during training in the UCSD-Ped1 dataset. To highlight curve variations, the y-axis for mAP@0.5 is set to a range of 0.9 to 1.0, while the y-axis for mAP@0.5:0.95 is set to a range of 0.5 to 1.0.

**Figure 8 sensors-25-02752-f008:**
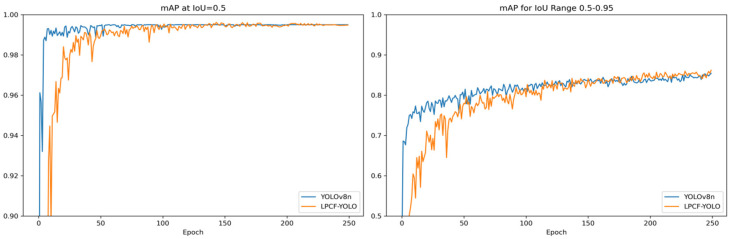
Visualization of mAP@0.5 and mAP@0.5:0.95 metrics during training in the UCSD-Ped2 dataset. To highlight curve variations, the y-axis for mAP@0.5 is set to a range of 0.9 to 1.0, while the y-axis for mAP@0.5:0.95 is set to a range of 0.5 to 1.0.

**Figure 9 sensors-25-02752-f009:**
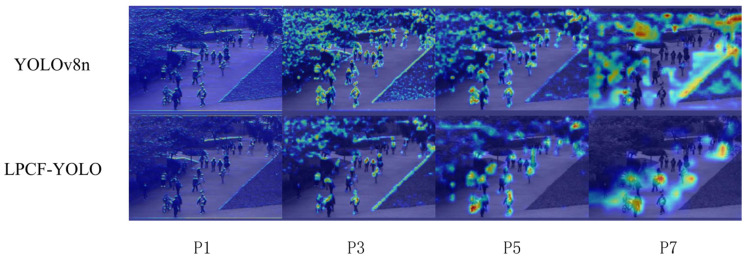
Comparison of heatmap results across different network layers between YOLOV8n and LPCF-YOLO.

**Table 1 sensors-25-02752-t001:** Label categories and quantities in the UCSD dataset.

UCSD	Label Category	Quantity
Ped1	Bike	2405
	Through the lawn	349
	Skateboard	1474
	Cart	685
	Wheelchair	328
	Pushcart	129
Ped2	Bike	1159
	Skateboard	290
	Cart	115

**Table 2 sensors-25-02752-t002:** Experimental Environment.

Item	Parameter
Operating System	Windows 10
Programming Language	Python 3.9
CPU	12thGen Intel(R) Core(TM) i9-12900H
GPU	NVIDIA RTX A2000 8 GB Laptop GPU
VRAM	8 GB
Algorithm Framework	PyTorch 2.5.0

**Table 3 sensors-25-02752-t003:** Performance comparison of different models in the UCSD-Ped1 dataset.

Method	Bike (%)	Skateboard (%)	Cart (%)	Wheelchair (%)	Through the Lawn (%)	Pushcart (%)	mAP @0.5 (%)	mAP @0.5:0.95(%)	Params/M	GFLO-Ps
YOLOv5n [20]	99.13	99.15	99.39	99.5	96.59	99.5	98.87	76.05	2.51	7.1
YOLOv7-tiny [21]	99.07	98.96	99.32	99.5	94.81	99.5	98.53	73.96	6.02	13.2
YOLOv8n [22]	99.41	99.08	99.38	99.5	95.17	99.5	98.67	75.74	3.00	8.1
YOLOv9t [23]	99.18	99.44	99.43	99.5	94.61	99.5	98.61	74.31	2.01	7.6
YOLOv10n [24]	99.20	99.22	99.43	99.5	95.81	99.5	98.76	75.12	2.70	8.2
YOLOv11n [25]	99.49	99.41	99.49	99.5	96.49	99.5	98.98	77.49	2.58	6.3
LPCF-YOLO	99.27	99.58	99.54	99.5	96.12	99.5	98.92	75.83	2.09	1.7

**Table 4 sensors-25-02752-t004:** Performance comparison of different models in the UCSD-Ped2 dataset.

Method	Bike (%)	Skateboard (%)	Cart (%)	mAP @0.5 (%)	mAP @0.5:0.95 (%)	Params/M	GFLOPs
YOLOv5n [20]	99.5	99.5	99.5	99.5	77.61	2.51	7.1
YOLOv7-tiny [21]	99.45	99.5	99.5	99.49	76.32	6.02	13.2
YOLOv8n [22]	99.49	99.5	99.5	99.5	76.85	3.01	8.1
YOLOv9t [23]	99.46	99.5	99.5	99.49	75.49	2.01	7.6
YOLOv10n [24]	99.48	99.5	99.5	99.49	76.53	2.7	8.2
YOLOv11n [25]	99.5	99.5	99.5	99.5	77.47	2.58	6.3
LPCF-YOLO	99.47	99.5	99.5	99.49	76.77	2.09	1.7

**Table 5 sensors-25-02752-t005:** The overall ablation experiment of LPCF-YOLO.

LPCF-YOLO-Backbone	L-HSFPN	WIOU	UCSD-Ped1 mAP @0.5(%)	UCSD-Ped2 mAP @0.5(%)	Params/M	GFLOPs	FPS
			98.67	99.5	3.00	8.1	15.7
√			98.71	99.48	3.19	2.1	44.3
	√		98.87	99.5	1.91	6.7	15.3
		√	98.65	99.5	3.00	8.1	15.7
√	√		98.64	99.5	2.09	1.7	42.9
	√	√	98.95	99.5	1.91	6.7	15.3
√		√	98.72	99.49	3.19	2.1	44.5
√	√	√	98.92	99.49	2.09	1.7	43.9

**Table 6 sensors-25-02752-t006:** Ablation experiment focused on the internal structure of the LPCF-YOLO-Backbone.

FPC-F	S-EMCP	ADown	UCSD-Ped1 mAP @0.5(%)	UCSD-Ped2 mAP @0.5(%)	Params/M	GFLOPs	FPS
			98.67	99.5	3.00	8.1	15.7
√			98.98	99.5	3.36	7.8	16.2
	√		98.78	99.5	2.84	8.0	15.9
		√	98.33	99.39	3.00	2.2	43.5
√	√		98.79	99.5	3.19	7.7	16.5
√	√	√	98.71	99.48	3.19	2.1	44.5

## Data Availability

No new data were created or analyzed in this study. Data sharing is not applicable to this article.

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
