# Peer review of "LPCF-YOLO: A YOLO-Based Lightweight Algorithm for Pedestrian Anomaly Detection with Parallel Cross-Fusion"

_sensors, 2025, doi:10.3390/s25092752_

Round 1

Reviewer 1 Report

Comments and Suggestions for Authors The paper introduces a lightweight pedestrian anomaly detection network—LPCF-YOLO—based on YOLOv8n. By incorporating FPC-F, S-EMCP, and ADown modules along with an improved L-HSFPN and the Wise-IoU loss function, the approach not only reduces the number of parameters and computational complexity but also maintains or slightly improves detection accuracy. Overall, the work is of high practical value for real-time detection in resource-constrained scenarios and effectively demonstrates the contributions of each module through comprehensive ablation studies. 1. The paper could further enhance the mathematical proofs and intuitive explanations regarding the design motivations and internal mechanisms of the FPC-F, S-EMCP, and ADown modules. 2. Some parts of the text are somewhat verbose; it is recommended to streamline the discussion to emphasize the key points. In addition, the annotations and explanations of figures and charts (e.g., module structure diagrams and heatmaps) could be more detailed to help readers better understand the design rationale behind each module and the underlying reasons for the experimental results. 3. Comparative experiments could be expanded—for example, by comparing LPCF-YOLO with other state-of-the-art lightweight detection models (not limited to the YOLO series)—to further illustrate its strengths and weaknesses across different scenarios. 4. The paper should discuss the detection performance under extreme lighting, low-resolution, and occlusion conditions, providing an objective analysis of the model’s limitations. 5. There are issues with the section numbering (e.g., sections labeled 4.51 and 4.52) that need to be corrected. 6. Regarding real-time performance, have the authors considered testing the model on edge devices? 7. It is recommended to include more discussion of detection methods in the background section, such as "Fast vehicle detection algorithm in traffic scene based on improved SSD" and "Automatic Potential Safety Hazard Evaluation System for Environment Around High-Speed Railroad Using Hybrid U-Shape Learning Architecture."

Author Response

Dear Reviewer,

We sincerely appreciate your constructive and insightful comments, which have significantly improved the quality of our manuscript. Your valuable suggestions regarding intuitive explanations, comparative experiments, and performance analysis have helped us refine the paper's clarity and technical rigor. We are particularly grateful for your guidance on streamlining the discussion and enhancing figure annotations, which has made our methodology more accessible to readers. These thoughtful recommendations have not only strengthened this work but also provided important directions for our future research in lightweight anomaly detection. Below is our point-by-point response to your feedback:

Comments 1: The paper could further enhance the mathematical proofs and intuitive explanations regarding the design motivations and internal mechanisms of the FPC-F, S-EMCP, and ADown modules.

Response 1:Considering that current deep networks are mostly designed empirically, it is difficult to provide a mathematical proof for the modules designed in this paper. However, intuitive explanations of the design motivation of FPC-F, S-EMCP, and ADown are added. For details, please refer to the introduction section of the paper. The internal working mechanisms have been specifically described in the methods section of the paper.

Comments 2:Some parts of the text are somewhat verbose; it is recommended to streamline the discussion to emphasize the key points. In addition, the annotations and explanations of figures and charts (e.g., module structure diagrams and heatmaps) could be more detailed to help readers better understand the design rationale behind each module and the underlying reasons for the experimental results.

Response 2:Some uninformative sentences have been deleted. The original Figure 2 also removed as it has been incorporated into Figure 1. In the introduction and methods sections of the paper, the design concept and some details of the module have been supplemented; in the experimental section of the paper, a specific cause analysis has been added for Figure 9  (Comparison of heatmap results across different network layers between YOLOV8n and LPCF-YOLO).

Comments 3: Comparative experiments could be expanded—for example, by comparing LPCF-YOLO with other state-of-the-art lightweight detection models (not limited to the YOLO series)—to further illustrate its strengths and weaknesses across different scenarios.

Response 3:This paper compares various lightweight versions of YOLOV5 - YOLOV11, which can basically represent state-of-the-art lightweight detection models. The number of parameters of object detection models based on Transformer (such as RT-DETR) is not in the same order of magnitude and they cannot be regarded as lightweight models, so no comparison is made.

Comments 4:  The paper should discuss the detection performance under extreme lighting, low-resolution, and occlusion conditions, providing an objective analysis of the model’s limitations.

Response 4:This dataset employed in this paper does not exhibit extreme lighting, low-resolution, and severe occlusion, so there are no experimental results in the experimental section. Considering the lack of corresponding training samples, the test results are not expected to be satisfied, and further optimization of the algorithm is needed. A brief explanation has been provided in the conclusion section of the paper.

Comments 5: There are issues with the section numbering (e.g., sections labeled 4.51 and 4.52) that need to be corrected.

Response 5:The numbering writing error has been corrected.

Comments 6: Regarding real-time performance, have the authors considered testing the model on edge devices?

Response 6:The algorithm in this paper has not yet been run on edge devices. The next step is to test and further optimize it on edge devices.

Comments 7:  It is recommended to include more discussion of detection methods in the background section, such as "Fast vehicle detection algorithm in traffic scene based on improved SSD" and "Automatic Potential Safety Hazard Evaluation System for Environment Around High-Speed Railroad Using Hybrid U-Shape Learning Architecture”

Response 7:Relevant literature[46]( "Fast vehicle detection algorithm in traffic scene based on improved SSD")[47]( "Fast vehicle detection algorithm in traffic scene based on improved SSD") has been supplemented: in the background section. â‘  “Single-stage detection methods directly regress the target's location and category, including approaches such as SSD (Single Shot MultiBox Detector)[19][46], and YOLO [19-24]”â‘¡ ” series [47] presented a novel hybrid learning architecture named UYOLO (U-shape You Only Look Once) which integrate the CSP-based backbone and detection branch to produce three scale high-level features for the object detection around the high-speed railroad.”

We believe that these revisions significantly enhance the quality and coherence of our manuscript. We are committed to making any further improvements that you may suggest.

Thank you for your consideration.

Best regards,

Authors

2024/4/15

Reviewer 2 Report

Comments and Suggestions for Authors

The authors propose a lightweight version of a pedestrian anomaly detection system. The topic addressed in the manuscript is timely and relevant, particularly considering the growing demand for efficient real-time anomaly detection in pedestrian monitoring applications. The authors' experiments demonstrate that, compared to YOLOv8n, the proposed method reduces the number of parameters and FLOPs while increasing FPS, maintaining or slightly improving detection accuracy. However, there are a number of minor comments that could help improve the quality of this manuscript.\
1. The authors should add a brief explanation of terms like "PConv", "SPPF", "C2F", and others in the introduction.
2. In Figure 1, the P5 output appears unused. The authors should clarify or correct this connection to ensure consistency in the architecture.
3. The LPCF-YOLO backbone structure is depicted redundantly in Figures 1 and 3. The authors should consider consolidating these representations to improve clarity.
4. The authors should clarify the behavior of the S-EMPC and ADown modules in Figure 4. Does the spatial dimension reduction occur after the AvgPool2d, AvgPool5d, and MaxPool5d layers? If so, please indicate the exact downsampling factors in Figure 2.
5. The authors should verify and correct the subscript notation in Equations (1), (4), and (5), as the current formatting appears unclear/illegible.
6. Equation (2) in the paper doesn’t match the corresponding equations in the original paper [Tong, Z., Chen, Y., Xu, Z., & Yu, R. (2023). Wise-IoU: Bounding box regression loss with dynamic focusing mechanism. arXiv 2023. arXiv preprint arXiv:2301.10051]. The authors should verify this equation and either align it with the reference or provide justification for any modifications.
7. The authors should add explanations for the variables and notations (r, β, δ, α, *, gt, x, y, W, H) in equations (1)-(6).

Author Response

Dear Reviewer,

We would like to express our sincere gratitude for your constructive feedback on our manuscript titled "LPCF-YOLO: A YOLO-Based Lightweight Algorithm for Pedestrian Anomaly Detection with Parallel Cross Fusion". We have carefully addressed each of your comments and below is our point-by-point response to your feedback:

Comments 1: The authors should add a brief explanation of terms like "PConv", "SPPF", "C2F", and others in the introduction.

Response 1:  Short explanations of terms such as 'PConv', 'SPPF', and 'C2F' have been added to the introduction section.

Comments 2:In Figure 1, the P5 output appears unused. The authors should clarify or correct this connection to ensure consistency in the architecture.

Response 2:Due to carelessness, one connecting line was missed at the original Figure P5. It has been added.

Comments 3: The LPCF-YOLO backbone structure is depicted redundantly in Figures 1 and 3. The authors should consider consolidating these representations to improve clarity.

Response 3:Figure 3 in the original paper has been deleted as it is indeed redundant.

Comments 4: The authors should clarify the behavior of the S-EMPC and ADown modules in Figure 4. Does the spatial dimension reduction occur after the AvgPool2d, AvgPool5d, and MaxPool5d layers? If so, please indicate the exact downsampling factors in Figure 2.

Response 4:Supplementary explanations have been provided. The first branch retains the original features without modification. The second branch applies 5×5 average pooling (stride 1, padding 2) followed by 5×5 max pooling(stride 1, padding 2) to capture contextual information. There is no spatial dimension reduction occur after the AvgPool2d, AvgPool5d, and MaxPool5d layers.

Comments 5: The authors should verify and correct the subscript notation in Equations (1), (4), and (5), as the current formatting appears unclear/illegible.

Response 5:The subscript notation has been verified . Since WIOUv4 is not the loss function proposed in this paper, a simplified treatment has been carried out, and each formula will no longer be introduced in detail.

Comments 6: Equation (2) in the paper doesn’t match the corresponding equations in the original paper [Tong, Z., Chen, Y., Xu, Z., & Yu, R. (2023). Wise-IoU: Bounding box regression loss with dynamic focusing mechanism. arXiv 2023. arXiv preprint arXiv:2301.10051]. The authors should verify this equation and either align it with the reference or provide justification for any modifications.  

Response 6:The corresponding equations has been verified to match the the original paper. Since WIOUv3 is not the loss function proposed in this paper, a simplified treatment has been carried out, and each formula will no longer be introduced in detail.

Comments 7: The authors should add explanations for the variables and notations (r, β, δ, α, *, gt, x, y, W, H) in equations (1)-(6).

Response 7:Since WIOUv3 is not the loss function proposed in this paper, a simplified treatment has been carried out, and each formula will no longer be introduced in detail.

We are committed to making any further improvements that you may suggest. Thank you once again for your time and effort in reviewing our work.

Best regards,

Authors

2024/4/16

Reviewer 3 Report

Comments and Suggestions for Authors

In this paper, the authors proposed a lightweight deep learning model for pedestrian anomaly detection. The topic is interesting, and the overall presentation of the paper is good. However, I have the following observations:

Although you claim that the model is lightweight compared to state-of-the-art models, there is no comparison table showing memory requirements, training time, number of trainable parameters, etc.

Author Response

Dear Reviewer,

We would like to express our sincere gratitude for your constructive feedback on our manuscript titled "LPCF-YOLO: A YOLO-Based Lightweight Algorithm for Pedestrian Anomaly Detection with Parallel Cross Fusion". We have carefully addressed each of your comments and below is our point-by-point response to your feedback:

Comments 1: In this paper, the authors proposed a lightweight deep learning model for pedestrian anomaly detection. The topic is interesting, and the overall presentation of the paper is good. However, I have the following observations: Although you claim that the model is lightweight compared to state-of-the-art models, there is no comparison table showing memory requirements, training time, number of trainable parameters, etc.

Response 1:Figures 7 and 8 show the learning curves, which illustrate the change of the loss function during the training process and the number of epochs required for convergence. Tables 3 and 4 present various experimental metrics, among which the number of model parameters can indicate the required memory.

Comments 2:Does the introduction provide sufficient background and include all relevant references?

Response 2:The introduction section provides an overview of the research background and includes important references. In this revision, two additional papers ([46][47]) have been added.

We are committed to making any further improvements that you may suggest. Thank you once again for your time and effort in reviewing our work.

Best regards,

Authors

2024/4/15

Reviewer 4 Report

Comments and Suggestions for Authors

The article presents LPCF-YOLO, a lightweight anomaly detection model based on YOLOv8n, aimed at reducing computational complexity for pedestrian anomaly detection. It introduces new modules—FPC-F, S-EMCP, ADown, and L-HSFPN—to replace parts of the original architecture, along with the Wise-IoU loss function to improve localization. Results on UCSD-Ped1 and Ped2 show reduced parameters and FLOPs, with improved FPS and comparable or slightly better mAP. The approach appears effective for real-time applications with limited resources. However, the following needs consideration in order to improve the quality of the manuscript:

  • An explicit novelty statement is missing from the introduction section. An explicit novelty statement in clear and concise manner help the readers understand the novel element of the research.
  • Table captions are not consistent throughout the article. Tables 1, 2, 5, and 6 have top captions, whereas Tables 3 and 4 have bottom captions. Table 6 caption needs to be on same page with the table itself.
  • The FPC-F module processes only 1/8 of the input channels through convolution to reduce FLOPs. What theoretical or empirical evidence is there that this level of sparsity will be sufficient to preserve representational richness, and how does performance degrade if this ratio is altered?
  • Given the FPC-F's inception from FasterNet and CSPNet, was the parameter saving versus semantic information saving trade-off mathematically estimated, specifically on small object detection tasks?
  • Given that big kernel pooling entails big receptive fields, have the authors addressed the risk of aliasing or over-smoothing in the pooled features? Can frequency-domain evidence be supplied to support their efficacy?
  • ADown introduces adaptive kernel-based padding. What mechanisms are there to preserve consistency and generalizability of learned padding behavior on different input resolutions and aspect ratios during inference?
  • WIoU v3 includes a gradient reweighting mechanism by outlier degree β. What is the behavior of this mechanism when there is label noise or class imbalance, and has it been demonstrated to introduce instability during initial training steps?
  • Equation (2) defines a non-monotonic IoU-based gradient modulation function. Does this function contain critical points or flat gradient regions that could affect convergence, particularly in borderline or ambiguous localization scenarios?
  • With a number of custom modules inducing additional branching (especially in neck and backbone), how is gradient flow synchronization managed in distributed or mixed-precision training?
  • Has there been a strong information-theoretic evaluation (e.g., preservation of mutual information or information bottleneck principle) conducted to justify the modular design choices in LPCF-YOLO?
  • Was LPCF-YOLO evaluated under post-training quantization (e.g., INT8, FP16) without fine-tuning? If so, how much is the degradation in performance, and what module contributes the most to quantization-induced loss?
  • In throughout the architecture there are several chains and concatenations used in parallel (e.g., in FPC-F and S-EMCP). Is the impact on memory bandwidth usage and cache consumption profiled for representative deployment hardware?
  • As the labeling was done manually using LabelImg and converted to YOLO format, what types of quality control or inter-annotator agreement checks were conducted to ensure labeling consistency between frames and anomaly types, especially for borderline cases like pedestrians on the lawn vs. normal pedestrians?
  • How did the authors address class imbalance, especially in Ped2 where some classes like 'Cart' have only 115 samples compared to 'Bike' with 1159? Was data augmentation, class re-weighting, or synthetic sample generation performed to prevent overfitting or model bias?
  • Since anomalies like skateboarding or lawn-crossing can be temporal in nature, how is the single-frame detection approach considering temporal cues? Would the inclusion of temporal information (e.g., by using 3D CNNs or LSTM layers) further improve detection accuracy, and was this considered?
  • LPCF-YOLO has only minor mAP improvements over some YOLO variants but with noticeably fewer FLOPs and parameters. For uses where high accuracy is the most important factor (e.g., monitoring sensitive areas), how do the authors defend the focus on light design at the cost of maximal accuracy?
  • All models were run with "default parameters"—how do the authors ensure that this does not bias performance comparison, especially since some of the YOLO variants may be underperforming simply due to suboptimal tuning to the UCSD dataset?
  • As mAP@0.5-0.95 dropped significantly compared to mAP@0.5, what anomaly types contributed most to this drop, and what does this reveal about LPCF-YOLO's localization robustness?
  • Though LPCF-YOLO reduces GFLOPs by a significant amount, were any real-time inference experiments conducted on edge devices (i.e., Jetson Nano or mobile GPU)? If so, what were the FPS and memory usage numbers, and where were the primary bottlenecks encountered?
  • The model was only evaluated on pre-defined anomaly types in UCSD. Can LPCF-YOLO generalize to new, unseen anomaly types (e.g., crawling, falling, or sudden running)? Zero-shot or open-set anomaly detection addressed or assessed?
  • Which specific modules in LPCF-YOLO (e.g., attention mechanisms, feature fusion layers) are most accountable for the GFLOPs reduction or the accuracy improvement? Component-wise ablation study conducted to understand this in detail?
  • Since the authors re-labeled the UCSD dataset in YOLO format and split it differently, is the entire labeled dataset (with annotations, splits, and preprocessing scripts) publicly available? If not, how will future researchers ensure reproducibility of the reported results?

Author Response

Dear Reviewer,

We would like to extend our sincere appreciation for your thorough and constructive evaluation of our manuscript, "LPCF-YOLO: A YOLO-Based Lightweight Algorithm for Pedestrian Anomaly Detection with Parallel Cross Fusion." Your comments have been extremely valuable in guiding our revisions and improving the overall quality of our work. We have carefully addressed each of your comments and below is our point-by-point response to your feedback:  

Comments 1: An explicit novelty statement is missing from the introduction section. An explicit statement in clear and concise manner help the readers understand the novel element of the research.

Response 1:The explanation of the novelty ideas of the paper has been added in the introduction part.

Comments 2:Table captions are not consistent throughout the article. Tables 1, 2, 5, and 6 have top captions, whereas Tables 3 and 4 have bottom captions. Table 6 caption needs to be on same page with the table itself.

Response 2:The format of all table captions in this paper have been checked and corrected

Comments 3: The FPC-F module processes only 1/ 8 of the input channels through convolution to reduce FLOPs. What theoretical or empirical evidence is there that this level of sparsity will be sufficient to preserve representational richness, and how does performance degrade if this ratio is altered?

Response 3:The input channels of convolutional layers often exhibit high redundancy. FPC-F module performs convolution operations on only a portion of the channels (such as 1/8) while retaining the original information of the remaining channels. This design is based on the trade-off between the precision and efficiency. If the channel processing ratio is increased (e.g., 1/4),it might introduce more redundant calculations. If decreased (e.g., 1/16), FLOPs further decrease, but the precision may drop due to information loss. As a hyperparameter, The channel processing ratio is selected based on experience.

Comments 4:  The FPC-F module processes only 1/8 of the input channels through convolution to reduce FLOPs. What theoretical or empirical evidence is there that this level of sparsity will be sufficient to preserve representational richness, and how does performance degrade if this ratio is altered?

Response 4:The trade-off between parameter preservation and semantic information preservation currently still relies on empirical judgment, lacking mathematical analysis and estimation.

Comments 5: The authors should verify and correct the subscript notation in Equations (1), (4), and (5), as the current formatting appears unclear/illegible.

Response 5:The subscript notation in Equations (1), (4), and (5) has been verified. Since WIOUv4 is not the loss function proposed in this paper, a simplified treatment has been carried out, and each formula will no longer be introduced in detail.

Comments 6:  Given the FPC-F's inception from FasterNet and CSPNet, was the parameter saving versus semantic information saving trade-off mathematically estimated, specifically on small object detection tasks?

Response 6: The trade-off between parameter preservation and semantic information preservation currently still relies on empirical judgment, lacking mathematical analysis and estimation.

Comments 7: Given that big kernel pooling entails big receptive fields, have the authors addressed the risk of aliasing or over-smoothing in the pooled features? Can frequency-domain evidence be supplied to support their efficacy?

Response 7: Pooling essentially is a downsampling operation. Using larger pooling kernels (such as 7×7) and larger strides can lead to the loss of high-frequency information in feature maps (e.g., edges, textures). If the downsampling rate (stride) does not satisfy the Nyquist-Shannon theorem with respect to the high-frequency components of the input features, high-frequency signals may be incorrectly folded into low frequencies, causing aliasing. Large kernel pooling, such as average pooling, significantly increases the receptive field, thereby covering larger areas of features. Its local weighted average can lead to blurring of details (such as loss of small objects), especially when the target size is smaller than the pooling kernel. Perform a Fast Fourier Transform (FFT) on the feature map and convert it into a spectral energy distribution. If the high-frequency energy after large kernel pooling is significantly reduced compared to small kernel pooling (such as by more than 30%), it indicates over-smoothing. If high-frequency energy does not decrease but unexplained noise appears in the frequency domain (such as high-frequency energy spreading to low frequencies), it indicates aliasing. This paper combines pooling layers with different kernel sizes (similar to Inception modules) to balance the receptive field and detail loss.

Comments 8: ADown introduces adaptive kernel-based padding. What mechanisms are there to preserve consistency and generalizability of learned padding behavior on different input resolutions and aspect ratios during inference?

Response 8: During training, the input is normalized to a base resolution (e.g., 224×224), and the parameters of the adaptive kernel are scaled proportionally to the actual input resolution. During inference, whether the input is 512×512 or 320×240, the kernel weights are adjusted through interpolation or predefined scaling rules to maintain consistent padding logic.

Comments 9:  WIoU v3 includes a gradient reweighting mechanism by outlier degree β. What is the behavior of this mechanism when there is label noise or class imbalance, and has it been demonstrated to introduce instability during initial training steps? Response 9: WIoU v3's gradient reweighting mechanism dynamically adjusts gradient gains based on the outlier degree β, which reflects anchor box quality. When label noise exists, highβ (lowβ) through its non-monotonic focusing coefficient, avoiding excessive bias toward either extreme. The gradient re-weighting mechanism achieves the suppression of harmful gradients and the focusing on effective samples in noisy and class-imbalanced scenarios through a dynamic non-monotonic strategy. Its design has been proven stable in standard detection tasks.

Comments 10: Equation (2) defines a non-monotonic IoU-based gradient modulation function. Does this function contain critical points or flat gradient regions that could affect convergence, particularly in borderline or ambiguous localization scenarios? Response10: The gradient modulation function of WIoU v3 indeed introduces non-monotonic characteristics. Its design may involve critical points or regions with flat gradients in scenarios such as boundary localization or ambiguous object positioning. However, its mechanism effectively mitigates potential convergence issues through dynamic adjustment strategies. The gradient modulation function of WIoU v3 uses a dynamic non-monotonic mechanism and intelligent gradient allocation strategy to introduce critical points while avoiding convergence stability issues. Its core design lies in adaptively adjusting gradient gain based on sample quality, rather than relying on fixed thresholds or monotonic penalties, thereby achieving more robust optimization in complex localization scenarios.

Comments 11: With a number of custom modules inducing additional branching (especially in neck and backbone), how is gradient flow synchronization managed in distributed or mixed-precision training?

Response11:  This study uses PyTorch for algorithm design and implementation. Gradient flow synchronization during training is not involved, and no abnormalities were observed during actual operation.

Comments 12: Has there been a strong information-theoretic evaluation (e.g., preservation of mutual information or information bottleneck principle) conducted to justify the modular design choices in LPCF-YOLO?

Response12:  Regarding the modular design of LPCF-YOLO, there is still a lack of information-theoretic analysis and evaluation. This may also be a common problem in the structural design of deep learning networks and is the direction for further efforts.

Comments 13: Was LPCF-YOLO evaluated under post-training quantization (e.g., INT8, FP16) without fine-tuning? If so, how much is the degradation in performance, and what module contributes the most to quantization-induced loss?

Response13:  The post-training quantization has not been discussed in this paper.

Comments 14:  In throughout the architecture there are several chains and concatenations used in parallel (e.g., in FPC-F and S-EMCP). Is the impact on memory bandwidth usage and cache consumption profiled for representative deployment hardware?

Response14:  This study systematically summarizes lightweight network design principles and computational metrics like FLOPs , hardware deployment optimization involving GPU memory bandwidth utilization, latency-throughput tradeoffs, and platform-specific adaptation strategies remains unexplored.

Comments 15: As the labeling was done manually using LabelImg and converted to YOLO format, what types of quality control or inter-annotator agreement checks were conducted to ensure labeling consistency between frames and anomaly types, especially for borderline cases like pedestrians on the lawn vs. normal pedestrians?

Response15: â‘ Standardization of predefined categories and annotation specifications: Clearly define all categories (e.g., "pedestrian_on_lawn" and "pedestrian") and establish detailed annotation rules.â‘¡ Coordinate visualization check: Use Python scripts to overlay annotated bounding boxes onto the original images to verify the correctness of coordinate normalization calculations. â‘¢Annotation tool-assisted verification: Utilize labelImg_annotation_checker to compare the results of different annotators.

Comments 16:  How did the authors address class imbalance, especially in Ped2 where some classes like 'Cart' have only 115 samples compared to 'Bike' with 1159? Was data augmentation, class re-weighting, or synthetic sample generation performed to prevent overfitting or model bias?

Response16:  For categories with a relatively small number of samples, this paper adopts data augmentation methods such as mirroring and rotation to address the class imbalance problem.

Comments 17:  Since anomalies like skateboarding or lawn-crossing can be temporal in nature, how is the single-frame detection approach considering temporal cues? Would the inclusion of temporal information (e.g., by using 3D CNNs or LSTM layers) further improve detection accuracy, and was this considered?

Response17: This paper adopts a single-frame detection method to detect anomalies without considering temporal sequence information. This is mainly based on lightweight design considerations. If 3D CNNs or LSTMs were used, the detection accuracy is expected to improve, but the network complexity would increase significantly.

Comments 18:  LPCF-YOLO has only minor mAP improvements over some YOLO variants but with noticeably fewer FLOPs and parameters. For uses where high accuracy is the most important factor (e.g., monitoring sensitive areas), how do the authors defend the focus on light design at the cost of maximal accuracy?

Response18: The method proposed in this paper is suitable for general detection scenarios. For applications where high accuracy is the most critical factor (e.g., monitoring sensitive areas), ensuring detection accuracy takes precedence, and lightweight design becomes relatively secondary. In such cases, adjustments to the network structure are needed.

Comments 19:  All models were run with "default parameters"—how do the authors ensure that this does not bias performance comparison, especially since some of the YOLO variants may be underperforming simply due to suboptimal tuning to the UCSD dataset?

Response19:  The various YOLO models compared with this paper were fine-tuned using the same training set and tested on the same test set, all based on default parameters, to ensure fairness in the comparison.

Comments 20:  As mAP@0.5-0.95 dropped significantly compared to mAP@0.5, what anomaly types contributed most to this drop, and what does this reveal about LPCF-YOLO's localization robustness?

Response20:  When the model's mAP@0.5-0.95 is significantly lower than mAP@0.5, it reflects deficiencies in the model's localization accuracy and strict detection requirements. mAP@0.5 only requires the Intersection over Union (IoU) between the predicted box and the ground truth box to reach 50% to be considered a correct detection, while mAP@0.5-0.95 demands that the model performs well across all IoU thresholds (from 0.5 to 0.95, in increments of 0.05). A decline in mAP@0.5-0.95 suggests that the model may not be suitable for tasks requiring high-precision localization, and this does not necessarily indicate poor detection performance for specific categories.

Comments 21: Though LPCF-YOLO reduces GFLOPs by a significant amount, were any real-time inference experiments conducted on edge devices (i.e., Jetson Nano or mobile GPU)? If so, what were the FPS and memory usage numbers, and where were the primary bottlenecks encountered?

Response21:  The method proposed in this paper has not yet been experimentally tested for real-time inference on edge devices (such as Jetson Nano) or mobile GPUs,which will be part of the next step of work.

Comments 22:  The model was only evaluated on pre-defined anomaly types in UCSD. Can LPCF-YOLO generalize to new, unseen anomaly types (e.g., crawling, falling, or sudden running)? Zero-shot or open-set anomaly detection addressed or assessed?

Response22: Considering the limitations of supervised learning, the model proposed in this paper still struggles to detect new types of anomalies without prior training.

Comments 23:  Which specific modules in LPCF-YOLO (e.g., attention mechanisms, feature fusion layers) are most accountable for the GFLOPs reduction or the accuracy improvement? Component-wise ablation study conducted to understand this in detail?

Response23: The experimental results on the impact of each module on mAP and GFLOPs have been presented in the ablation experiments (Table 6) in Section 4.5.2 of this paper, and a detailed analysis of the experimental data has been conducted.

Comments 24:  Since the authors re-labeled the UCSD dataset in YOLO format and split it differently, is the entire labeled dataset (with annotations, splits, and preprocessing scripts) publicly available? If not, how will future researchers ensure reproducibility of the reported results?

Response24:  If the paper is published, the entire labeled dataset and related source code will be made publicly available in a timely manner.

We believe that these revisions address your concerns comprehensively and significantly enhance the quality of our manuscript. We are committed to making any further improvements as suggested.

Thank you once again for your valuable feedback and the opportunity to improve our work. We look forward to your further comments and the possibility of contributing to your esteemed journal.

Best regards,

Authors

2024/4/15

Round 2

Reviewer 1 Report

Comments and Suggestions for Authors

The author addressed my concerns. However, there are some problems with the format of the thesis literature, especially the author's name in Literature 47, which needs to be carefully checked

Author Response

Dear reviewer,

We would like to express our sincere gratitude for your constructive feedback on our manuscript . We have carefully addressed your comments and below is our point-by-point response to your feedback:

Comments :The author addressed my concerns. However, there are some problems with the format of the thesis literature, especially the author's name in Literature 47, which needs to be carefully checked

Response: All references, including reference [47], have been checked and corrected for format. The modified parts are displayed in green font.

Best regards

Authors

2025/4/20

Reviewer 2 Report

Comments and Suggestions for Authors

The all comments are addressed adequately.
The authors should pay attention to the text marked in red. For example, the verb in line 338, "α Controls", should start with a lowercase letter. Additionally, the phrase "c threshold" in line 339 is unclear.
Also, the authors should carefully proofread the article and correct the formatting. For example, it is necessary to add paragraph indents before sections 3.1, 3.2, 4.5, 4.5.2, and 4.5.3. Also, pay attention to the left margins of Table 3.

Author Response

Dear reviewer,

We would like to express our sincere gratitude for your constructive feedback on our manuscript . We have carefully addressed your comments and below is our point-by-point response to your feedback:

Comments:The all comments are addressed adequately. The authors should pay attention to the text marked in red. For example, the verb in line 338, "α Controls", should start with a lowercase letter. Additionally, the phrase "c threshold" in line 339 is unclear. Also, the authors should carefully proofread the article and correct the formatting. For example, it is necessary to add paragraph indents before sections 3.1, 3.2, 4.5, 4.5.2, and 4.5.3. Also, pay attention to the left margins of Table 3.

Response: A comprehensive check has been carried out on the revised paragraphs, and some details with imperfect expressions have been modified. The sentence in line 338 has been revised to “The parameter α controls the curvature of the focusing mechanism. A value of 1.8 balances attention between common and outlier samples. Set δ to typical value 3.“; Paragraph indents have added before sections 3.1, 3.2, 4.5, 4.5.2, and 4.5.3. The left margins of Table 3 has been adjusted. The modified parts are displayed in green font.

 Best regards

Authors

2025/4/19

Reviewer 4 Report

Comments and Suggestions for Authors

The authors have addressed all the comments.

Author Response

Dear reviewer,

We would like to express our sincere gratitude for your constructive feedback on our manuscript . We have carefully addressed your comments and below is our response to your feedback:

Comments:The authors have addressed all the comments.

Response: Thank you for your review. A comprehensive check has been carried out on the revised paragraphs, and some details with imperfect expressions have been modified. The modified parts are displayed in green font.

Best regards

Authors

2025/4/20